# Impact of a Video-Based Educational Intervention on the Levels of Knowledge and Concerns about COVID-19 Vaccination

**DOI:** 10.3390/vaccines11040727

**Published:** 2023-03-24

**Authors:** Najla A. Barnawi, Basmah Alraqei, Ashwaq Hilwan, Maram Al-Otibi, Roaya Alsubaie, Shahad Altowymy, Mostafa A. Abolfotouh

**Affiliations:** 1College of Nursing, King Saud Bin Abdulaziz University for Health Sciences, Ministry of National Guard, Riyadh 11426, Saudi Arabia; 2King Abdullah International Medical Research Center (KAIMRC), King Saud Bin Abdulaziz University for Health Sciences, Ministry of National Guard, Riyadh 11426, Saudi Arabia

**Keywords:** COVID-19 vaccination, concerns, knowledge, vaccine hesitancy, vaccination acceptance, e-learning, Saudi Arabia

## Abstract

Background. The hesitancy to receive the COVID-19 vaccine plays a role in delaying the current global and national COVID-19 management strategies. Evidence has highlighted the importance of examining the public’s concerns and knowledge about COVID-19 vaccines in sustaining public prevention of the further spread of the virus worldwide. This study aimed to assess the impact of a video-based educational session on the Saudi public’s levels of knowledge and concerns about the COVID-19 vaccination. Methods. In a double-blind, randomized posttest-only control group study design, 508 Saudis were randomized to an experimental group (n = 253) and a control group (n = 255). The experimental group was exposed to a video-based educational session, while the control group was not. Then both groups were subjected to a validated questionnaire to assess their level of knowledge and concerns about the vaccine. Results. In comparison to the control group, the experimental group showed a significantly lower proportion of overall high concern (0.4% vs. 5.5%, *p* < 0.001) and a higher proportion of overall good knowledge (74.2% vs. 55.7%, *p* < 0.001). After adjustment for possible confounders, the experimental group showed a significantly lower percent mean score of overall concern (45.0% vs. 65.0%, *p* < 0.001) and a higher percent mean score of overall knowledge (74.2% vs. 55.7%, *p* < 0.001) than the control group. Conclusions. The video-based educational intervention positively impacted the levels of knowledge and concerns about COVID-19 vaccination among the experimental group. These interventions safeguard against the rumors and misconceptions about COVID-19 vaccinations. Further studies on the impacts of such interventions on vaccine uptake are recommended.

## 1. Introduction

Managing the COVID-19 pandemic through sustainable and effective global preventive measures has played a crucial role in controlling the massive spread of the virus worldwide. Furnishing the public with COVID-19 vaccines nationally and internationally sustains the intended aim of the preventive measures [1,2]. Despite the success of those strategies in controlling the spread of the virus, evidence indicates some public hesitancy about the effectiveness of the COVID-19 vaccines [1,2,3,4]. Several concerns have led to this hesitancy, such as worries about the vaccine’s risks and safety, mistrust, and concerns about commercial profiteering at the expense of human health. According to Lazarus et al. [5], 71.5% of 13,426 individuals from different countries were concerned about the vaccine’s safety and effectiveness. They also highlighted that one’s willingness to be vaccinated did not indicate acceptance and trust in COVID-19 vaccines [5]. Fridman et al. [6] made a similar claim. They acknowledged the impact of a lack of knowledge about the disease and the current vaccines on public hesitancy. Batteus et al. [7] reported that lack of knowledge is one barrier that increases public concerns, leading to hesitancy to receive COVID-19 vaccines.

It is essential to address the association of public concerns and hesitancy about COVID-19 vaccines with demographic factors such as gender and age, the existence of rumors and conflicting media coverage, and the overlapping vaccine-related socio-cultural and political ideologies. El-Elimat et al. [8] investigated the acceptability of receiving any of the COVID-19 vaccines among 3100 Jordanians and its association with factors that impact their attitudes. Age was one factor associated with the lack of knowledge about COVID-19, which decreased the participants’ willingness to receive the vaccine, mainly among those who were 35 years and older. Qunaibi et al. [9] reported that female, Arabic-speaking participants, aged 30–59 years and with confirmed cases of COVID-19, lacked knowledge regarding the types of vaccines. Participants who had not received any influenza vaccine showed significantly higher vaccine hesitancy.

Understanding the barriers to COVID-19 vaccine uptake is essential. However, there is a need to examine the effectiveness of experimental interventions concentrating on COVID-19 vaccine barriers, such as public concerns about the vaccine and their intentions of receiving it. Several interventions have popped up in the public health field, such as communication campaigns, incentivized public strategies, and reminder applications and software [7]. Globally, since the pandemic’s beginning, 32 interventional studies with various experimental designs have examined their impacts on the population’s intentions toward vaccine uptake [7,10,11,12].

Saudi Arabia (SA) is a pioneer in the Arab world and has taken a prompt and effective lead in managing COVID-19. The Saudi government adopted the World Health Organization’s (WHO) COVID-19 guidelines to manage the pandemic based on nine pillars of public health preparedness and response [13]. Some pillars that have had a compelling influence in managing the pandemic were knowledge awareness campaigns and advanced research in COVID-19 vaccines—responding to the national pandemic strategies and illustrating Saudi Vision 2030, which focuses on integrating digital transformation and artificial intelligence. In addition, various national studies were conducted to address the concerns about and patterns of COVID-19 vaccination in the Saudi context [14,15,16].

Numerous studies have highlighted the positive impact of reducing the population’s hesitancy and concern about COVID-19 vaccination uptake [5,17,18,19,20,21]. Others have highlighted the importance of integrating multifaceted educational and awareness interventions that focus on the efficacy and safety of the vaccines, which enhanced the Saudi population’s acceptance of COVID-19 vaccination uptake [16,22]. Thus, this study aimed to assess the impact of a video-based educational session on the Saudi public’s knowledge and concerns about the COVID-19 vaccination. Further, this study aimed to assess the impact of a video-based educational session, in comparison to traditional learning from the national health authorities, on the Saudi public’s levels of knowledge and concerns about the COVID-19 vaccination.

## 2. Methods

### 2.1. Study Design

This study was a posttest-only control group experimental design conducted to assess the impact of a video-based intervention on the participants’ knowledge and concerns about receiving the COVID-19 vaccine. To maintain the physical distancing strategy and preventive measures, we developed an online questionnaire via Survey Monkey (https://www.surveymonkey.com/r/8K688DH) (accessed on 20 September 2021) restricted to one participation per unique internet protocol (IP) address. Two links were developed: one for the experimental group included the video-based educational intervention and the posttest questionnaire, while the other for the control group had only the questionnaire.

### 2.2. Study Participants

We utilized the random selection option on the SurveyMonkey collecting data webpage on 12 October 2021, which allowed us to send both links randomly to various Saudi populations. The inclusion criteria were added to SurveyMonkey; these were Saudi citizens who could read and communicate in Arabic or English and were aged ≥ 18 years at the study date. People who engaged in any COVID-19 vaccine campaigns were automatically excluded, along with those who lacked Internet access or could not complete an online survey for any reason. Figure 1 is a flow diagram of the experimental and control group allocations.

Based upon a 50% incidence of vaccination concern [23] for the control group and 35% for the experimental group (with a 15% post-intervention reduction in the proportion of concerned individuals), with α 0.05, β 0.10, and power 0.90, an estimated sample size of 242 individuals was required in each group. Thus, 484 individuals in the experimental and control groups comprised the approximate number needed to demonstrate statistical significance. The number of participants reached 500, divided equally into 250 for each group.

Participation in this study was voluntary; the participants were asked if they agreed to participate and were assured their responses would remain anonymous. Each participant received an electronic informed consent form through the management system, which had a short description of the study and asked potential subjects for their consent to participate. The study protocol was approved by the Institutional Review Board (IRB) of the Ministry of National Guard–Health Affairs (MNGHA), Riyadh, Saudi Arabia (Protocol # NRC21R/420/10). This study was conducted following the Declaration of Helsinki.

### 2.3. Educational Intervention

This study examined the intervention, including two videos about the COVID-19 vaccines. The first video contained clips of the types, components, and mechanisms of approved COVID-19 vaccines in Saudi Arabia (5 min. duration). The second video covered COVID-19 vaccine-related rumors and the disclaimers for each of these rumors based on scientific evidence (10 min. duration). The video materials and content were retrieved from the WHO (2020) [24], CDC (2021) [25], Saudi Ministry of Health (2021a) [26], and the COVID-19 Campaign Center at National Guard Health Affairs [27]. The content was delivered in Arabic, and English translation captions were included in the videos. An online debriefing page included a general overview of the study, its purpose, a consent form, and instructions on using the module. The online, self-administered posttest questionnaire was presented to participants of both groups once the experimental group received the study intervention.

### 2.4. Data Collection Methods and Outcome Measures

Assessment of Knowledge about vaccination:

The COVID-19 knowledge questionnaire had 19 close-ended questions based on the COVID-19 video-educational intervention content. Each question was responded to with “True” (1 point) or “False, or Don’t know” (0 points). Knowledge was assessed in four domains: vaccine mechanisms, the formation process, rumors about vaccination, and COVID-19 vaccines available in Saudi Arabia. Total and percentage knowledge scores were calculated for each participant. The levels of knowledge were categorized into three categories: good (>75% score), average (50–75% score), and poor (<50% score).

Assessment of concern about COVID-19 vaccination:

The vaccination attitude examination (VAX) five-point Likert scale [28] was used to assess the participants’ concerns about COVID vaccination. It included 12 items that were classified into vaccine perceptions with four domains: mistrust of vaccine benefits, worries about the effects, commercial profiteering concerns, and natural immunity. The total concern score ranged from 12 to 60 points, where 12 indicated the lowest score and 60 indicated the highest score. Total and percentage mean scores were calculated for each of the four domains. The levels of concern were categorized into three categories: high (>75% score), average (50–75% score), and low (<50% score).

Personal and COVID-19-related characteristics:

The investigators developed a set of 14 questions that covered general demographic data, COVID-19-related characteristics such as any current symptoms of COVID-19, working in any medical field, history of allergy of any kind, phobia of needles, and taking drugs such as anti-inflammatory, anticoagulant, immune-suppressive, or antipsychotic medications.

The study intervention was piloted with a lay Saudi population of 40, who were not included in this study but agreed on its quality, practicality, accessibility, and audiovisual materials.

### 2.5. Data Analysis

Data entry and statistical analysis were performed with the statistical package for the social science (SPSS) software program for Windows (version 28.0.1.1, © copyright IBM corporation, Armonk, NY, USA). Descriptive statistics were calculated, such as percentages, means, and standard deviations. The chi-square test was applied for categorical data, and student-independent t tests and ANOVA were used for continuous data. Multiple regression analyses were performed to control for the following independent variables in the experimental and control groups regarding the levels of concerns and knowledge in relation to their different domains: study group (experimental group), gender, and taking an anti-inflammatory and an immunosuppressive medication. Statistical significance was considered at *p* < 0.05 for all analyses.

### 2.6. Ethical Considerations

The study was approved by the Institutional Review Board (IRB) of the Ministry of National Guard–Health Affairs (MNGHA), study number NRC21R/420/10. All methods were carried out following relevant guidelines and regulations.

## 3. Results

### 3.1. Personal and COVID-19-Related Characteristics

Table 1 shows the personal characteristics and COVID-19-related characteristics for both the experimental and control groups. The experimental group was significantly different from the control group in gender (χ^2^ = 8.95, *p* = 0.003), history of taking anti-inflammatory drugs (24.5% vs. 4.3%, *p* < 0.001) and history of taking immune-suppressive drugs (13.4% vs. 7.8%, *p* = 0.041).

### 3.2. Knowledge about COVID-19 Vaccination

Table 2 shows the levels of knowledge about the COVID-19 vaccination in the experimental and control groups. In comparison with the control group, the experimental group showed a significantly high proportion of good knowledge of the general vaccine mechanism (87.4% vs. 43.5%, *p* < 0.001), information about vaccines (73.1% vs. 35.7%, *p* < 0.001), vaccine formation process (7.9% vs. 5.9%, *p* < 0.001), availability of COVID-19 vaccine in Saudi Arabia (28.5% vs. 7.8%, *p* < 0.001), and overall knowledge (71.9% vs. 4.3%, *p* < 0.001). The experimental group showed a significantly lower percentage mean score on the general vaccine mechanism (90.8% vs. 68.2%, *p* < 0.001), rumors about vaccines (82.1% vs. 53.2%, *p* < 0.001), the vaccine formation process (54.3% vs. 43.1%, *p* < 0.001), COVID-19 vaccine availability in Saudi Arabia (62.2% vs. 51.1%, *p* < 0.001), and overall knowledge (79.2% vs. 55.7%, *p* < 0.001).

Figure 2 shows the percentage mean score of the knowledge domains about COVID-19 vaccination in the experimental and control groups. The impact of the educational intervention was the highest on rumors, followed by the general vaccine mechanism, the vaccine formation process, and the availability of vaccines in Saudi Arabia.

#### Concerns about COVID-19 Vaccination

Table 3 shows the levels of concern about the COVID-19 vaccination in the experimental and control groups. In comparison with the control group, the experimental group showed a significantly lower proportion of high concern in the mistrust domain (2.8% vs. 14.5%, *p* < 0.001), worries domain (1.2% vs. 11.4%, *p* < 0.001), commercial profiteering domain (0.0% vs. 22.4%, *p* < 0.001), preference for natural immunity (5.5% vs. 12.9%, *p* < 0.001), and overall concern (0.4% vs. 5.5%, *p* < 0.001). The experimental group showed a significantly lower percentage mean score of concern than the control group on mistrust (38.1% vs. 66.4%, *p* < 0.001), worries (59.2% vs. 63.6%, *p* < 0.001), commercial profiteering (44.0% vs. 66.2%, *p* < 0.001), and overall concern (45.0% vs. 65.0%, *p* < 0.001).

Figure 3 shows the percentage mean scores of concerns about COVID-19 vaccination in the experimental and control groups. The impact of the educational intervention on concern about COVID-19 vaccination was highest in the mistrust domain, followed by the natural immunity, commercial profiteering, and worries domains.

Table 4 shows the impact of the educational intervention on the overall knowledge and the overall concern about the COVID-19 vaccination after adjusting for gender, use of anti-inflammatory drugs, and use of immune-suppressive drugs. It indicated a significant positive association between the intervention and the percent mean score of the overall knowledge (t = 10.24, *p* < 0.001), reflecting a significant positive impact of the intervention on the experimental group. Meanwhile, a significant negative association between the intervention and the percentage mean score of the overall concern (t = −27.00, *p* < 0.001) reflected a significant positive impact of the intervention on the experimental group. A history of anti-inflammatory drug use was associated with a significantly higher percentage mean score of overall knowledge about COVID-19 vaccination (t = 2.16, *p* = 0.03), while the history of immune-suppressive drug use was associated with a significantly lower percentage mean score of overall concern about COVID-19 vaccination (t = −2.76, *p* = 0.006).

## 4. Discussion

Examining the concerns and knowledge about receiving COVID-19 vaccines among the Saudi population is essential in healthcare. Several national studies examined the association of concerns and knowledge about vaccination with the hesitancy to receive COVID-19 vaccines [29,30,31,32,33,34]. Still, the current study is the first of its kind in Saudi Arabia that takes an interventional approach to study the impact of an educational video on the knowledge and concerns about COVID-19 vaccination. Our study showed a positive effect of the video intervention on raising the levels of knowledge and lowering the concerns about COVID-19 vaccination among the participants in the experimental group, which may enhance the intention of receiving the vaccine.

Numerous studies have shown the effectiveness of an intervention-based approach in minimizing the concerns and enhancing the COVID-19 vaccination uptake [35,36]. Dai et al. [35] highlighted that adding educational videos to vaccination appointment reminders increased their effectiveness and the participant’s likelihood of vaccination. In our study, the experimental group showed significantly lower percentage mean scores in all concern domains than the control group. Moreover, the impact of educational intervention on concern about the COVID-19 vaccination was the highest in the mistrust domain. In a previous study on recovered COVID-19 patients, mistrust was the main attitudinal barrier to receiving the vaccine; the majority of participants were not sure that vaccination would stop them from contracting serious infectious diseases [37]. However, including clips that were structured specifically to focus on COVID-19 vaccine-related rumors and the scientific disclaimers of each of these rumors based on evidence would justify such an impact. In Saudi Arabia, collaborative efforts between the Saudi Center for Disease Control and the Ministry of Health were made to provide effective educational campaigns supported by various EduTech initiatives [13]. This reflected the importance of integrating highly advanced education-based sessions about COVID-19 vaccines.

Intervening in the cultural, social, and religious misconceptions and rumors about COVID-19, mainly in unregulated social media, negatively impacts knowledge and increases population concerns about receiving COVID-19 vaccines [29,38,39,40,41]. Our study indicated a positive impact on improving the knowledge of COVID-19 among participants in the experimental group. Moreover, the impact of educational intervention was the highest on rumors. Again, this impact could be attributed to the use in our study of clips that were specially structured to cover COVID-19 vaccine-related rumors and the scientific disclaimers of each of these rumors based on evidence. Accessible information on the safety of COVID-19 vaccines accelerated the vaccination process and minimized the hesitancy and concerns about vaccinations within the Saudi context [42]. These results reflected the importance of providing accurate knowledge about the COVID-19 vaccines, which highlights the effect of providing reliable information that focuses on the types, safety, and mechanisms.

The influence of receiving certain medications, such as anti-inflammatory, immunosuppressive, and antipsychotic medications, on COVID-19 outcomes has been reported [43,44]. In our study, both histories of taking anti-inflammatory drugs and immune suppressive drugs impacted positively on the overall knowledge and concern about the COVID-19 vaccination by raising the levels of knowledge and lowering the levels of concern about vaccination among drug takers. This finding could be justified by the fact that participants on multiple drug therapy had more concern and hesitance about taking the COVID-19 vaccine as it might deteriorate the prognosis of their condition [43,45]. Ren et al. [29] claimed that psychiatric patients and their family caregivers in China agreed to receive vaccination and had a greater likelihood of joining the COVID-19 immunization program if they knew more about the COVID-19 vaccine coverage and safety, which decreased their concerns [44].

In a previous study on the Saudi public, female sex was a significant predictor of compliance to precautionary measures, preparedness, self-quarantine activities, and overall behavior [4]. Women are usually able to break with tradition to safeguard the health of their children and adjust the community’s system to their best advantage [46]. However, in the present study, after adjusting for possible confounders, there was no sex difference in the impact of the video-based educational intervention.

### Strengths and Limitations

This study used a randomized posttest-only approach to evaluate the impact of the video-based intervention on the experimental group compared to the control group, a design that minimized the threats to the internal validity of our conclusion, such as history, maturation, testing, mortality, and others. It also allowed generalization of the study’s conclusion. Furthermore, this study enhanced the concept of e-learning via the telehealth approach, considered a vital recommended strategy to overcome the misconceptions and myths about the COVID-19 vaccine. Such an approach is compatible with Saudi Vision 2030 and global strategies that aim to reinforce the applicability of utilizing a safe and distance model of care during pandemic periods [47].

However, this study has limitations. One of these is that the posttest was administered immediately after the video. Although this methodology tested the amount of knowledge and concern acquired after the intervention session, it failed to evaluate the long-term retention of such outcomes. Another limitation was the absence of pre-intervention data to compare with the post intervention data. Moreover, this study examined the impact of the educational intervention on changes in levels of knowledge and concerns about the vaccines but did not examine its impact on actual vaccine uptake; further studies are recommended.

## 5. Conclusions

Our results indicated that the video-based educational interventions significantly increased knowledge about COVID-19, its mechanisms, its vaccines, and its rumors. It is essential to reinforce public awareness within the national health policies to reconstruct the necessary and up-to-date public knowledge about COVID-19 and its associated vaccines. Thus, it will enhance the Saudi population’s attitudes toward appropriate preventive measures in similar pandemics. For instance, developing a national task force that includes several experts in pandemics, health education, and public health is essential to regaining the appropriate awareness of COVID-19 or other pandemics. Integrating large-scale, self-assessed educational interventions supported by national social media in collaboration with the Ministry of Health (MOH) is required to enhance the transparent transformation of information regarding pandemic management strategies for the Saudi public.

## Figures and Tables

**Figure 1 vaccines-11-00727-f001:**
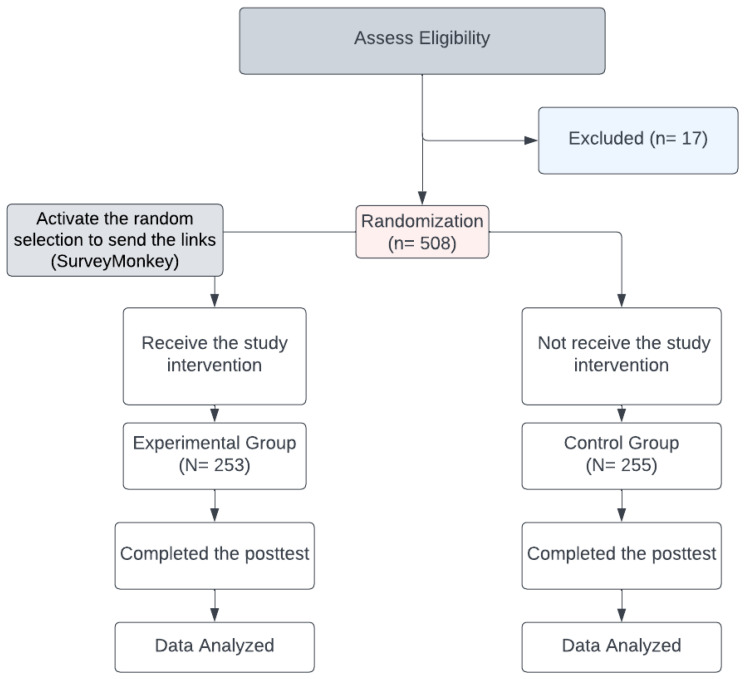
Flow diagram of the allocation of experimental and control groups.

**Figure 2 vaccines-11-00727-f002:**
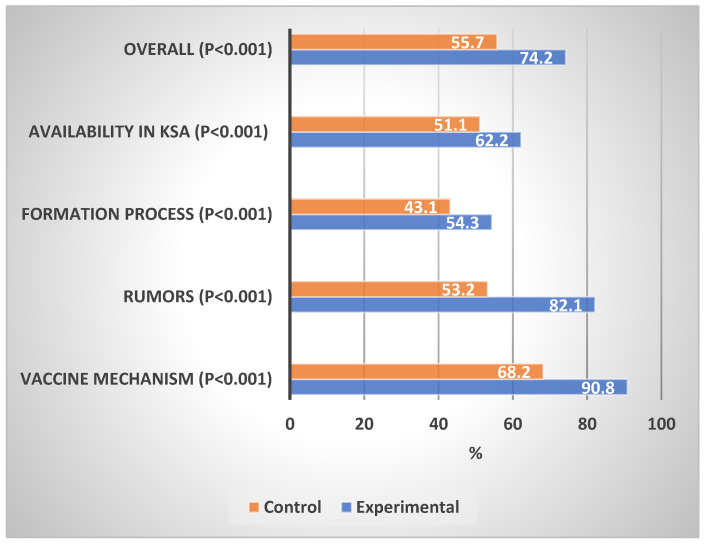
Percentage mean scores of the knowledge about COVID-19 vaccines. Differences in Experimental vs. Control Groups.

**Figure 3 vaccines-11-00727-f003:**
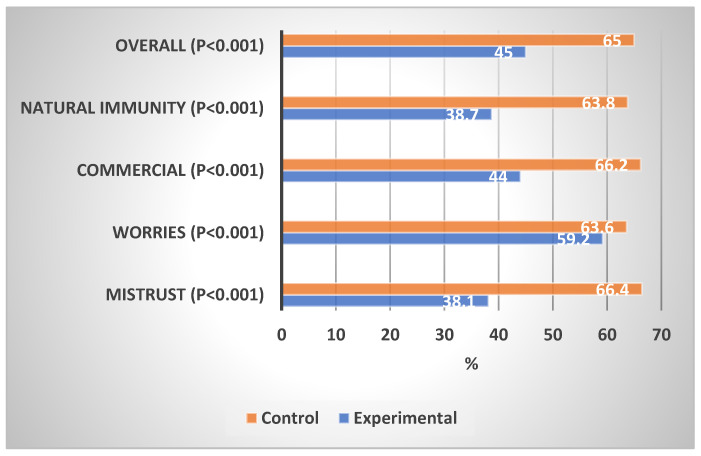
Percentage mean scores of COVID-19 concerns of the experimental and control groups.

**Table 1 vaccines-11-00727-t001:** Personal and COVID-19-related characteristics of the experimental and control groups.

Characteristics	Experimental Group		Control Group		χ^2^	*p*-Value
	No (253)	%	No (255)	%		
Personal characteristics						
Gender						
Female	113	44.7	81	31.8	8.95	0.003 *
Male	140	55.3	174	68.2		
Age						
Less than 45 years	222	87.7	222	87.1	0.06	0.82
45 years or more	31	12.3	33	12.9		
Education						
Secondary or less	65	25.7	70	27.5	0.20	0.65
University or above	188	74.3	185	72.5		
Marital Status						
Single	120	47.4	114	44.9	5.77	0.12
Married	97	38.3	119	46.7		
Widowed	5	2.0	3	1.2		
Divorced	31	12.3	19	7.5		
Monthly Income						
Low	88	34.8	68	26.7	4.25	0.12
Medium	132	52.2	145	56.9		
High	33	13.0	42	16.5		
Pregnancy Status (females only)						
Yes	18	37.5	12	22.2	2.86	0.09
No	30	62.5	42	77.8		
Working in Medical Field						
Yes	78	30.8	92	36.1	1.57	0.21
No	175	69.2	163	63.9		
COVID-19-related characteristics						
COVID-19 Symptoms						
Yes	55	21.7	48	18.8	0.67	0.41
No	198	78.3	207	81.2		
Allergies to COVID-19 Vaccines						
Yes	37	14.6	37	14.5	0.001	0.97
No	216	85.4	218	85.5		
Phobia of Needles						
Yes	34	13.4	34	13.3	0.001	0.97
No	219	86.6	221	86.7		
Taking Any Anti-inflammatory						
Yes	62	24.5	11	4.3	42.08	<0.001 *
No	191	75.5	244	95.7		
Taking Any Blood Anticoagulants						
Yes	23	9.1	14	5.5	2.44	0.12
No	230	90.9	241	94.5		
Taking Any Immune-suppressive						
Yes	34	13.4	20	7.8	4.19	0.041 *
No	219	86.6	235	92.2		
Taking Any Antipsychotics						
Yes	32	12.6	23	9.0	1.73	0.19
No	221	87.4	232	91.0		

* Statistical significance at *p* ≤ 0.05, χ^2^—Pearson chi-square test.

**Table 2 vaccines-11-00727-t002:** Levels of knowledge about the COVID-19 vaccination in different domains of the experimental and control groups.

Level of Knowledge											
Knowledge Domains	Poor (Less than 50%)		Average (50–75%)		Good (More than 75%)		χ^2^	*p*-Value	PMS (SD)	t-Value	*p*-Value
	N	%	N	%	N	%					
Vaccine Mechanisms											
Control Group	83	32.5	61	23.9	111	43.5	113.68	<0.001	68.2 ± 24.0	−11.35 *	<0.001
Experimental Group	28	11.1	4	1.6	221	87.4			90.8 ± 21.0		
Rumors											
Control Group	144	56.5	20	7.8	91	35.7	85.56	<0.001	53.2 ± 38.5	−8.92 *	<0.001
Experimental Group	44	17.4	24	9.5	185	73.1			82.1 ± 34.3		
Vaccine Formation Process											
Control Group	145	56.9	95	37.3	15	5.9	48.72	<0.001	43.1 ± 28.3	−4.37 *	<0.001
Experimental Group	67	26.5	166	65.6	20	7.9			54.3 ± 29.2		
COVID-19 Vaccines in Saudi Arabia											
Control Group	180	70.6	55	21.6	20	7.8	70.72	<0.001	51.1 ± 17.1	−6.30 *	<0.001
Experimental Group	88	34.8	93	36.8	72	28.5			62.2 ± 22.2		
Overall Knowledge											
Control Group	82	32.2	162	63.5	11	4.3	249.26	<0.001	55.7 ± 14.3	−11.59 *	<0.001
Experimental Group	33	13.0	38	15.1	182	71.9			74.2 ± 21.0		

χ^2^—Pearson Chi square test, *—statistical significance.

**Table 3 vaccines-11-00727-t003:** Levels of concern about COVID-19 vaccination in different domains of the experimental and control groups.

Level of Concern											
Concern Domain	Low (Less than 50%)		Moderate (50–75%)		High (More than 75%)		χ^2^	*p*-Value	PMS (SD)	t-Value	*p*-Value
	No	%	No	%	No	%					
Mistrust											
Control Group	15	5.9	203	79.6	37	14.5	306.14	< 0.001	66.4 ± 15.1	24.76 *	<0.001
Experimental Group	210	83.0	36	14.2	7	2.8			38.1 ± 10.0		
Worries											
Control Group	33	12.9	193	75.7	29	11.4	22.65	<0.001	63.6 ±11.1	5.18 *	<0.001
Experimental Group	32	12.6	218	86.2	3	1.2			59.2 ± 7.7		
Commercial Profiteering											
Control Group	33	12.9	165	64.7	57	22.4	192.19	<0.001	66.2 ± 13.7	19.93 *	<0.001
Experimental Group	179	70.8	74	29.2	0	0.0%			44.0 ± 11.3		
Preference for Natural Immunity											
Control Group	43	16.9	179	70.2	33	12.9	191.42	<0.001	63.8 ± 13.0	18.11 *	<0.001
Experimental Group	197	77.9	42	16.6	14	5.5			38.7 ± 17.9		
Overall Concern											
Control Group	1	0.4	240	94.1	14	5.5	313.44	<0.001	65.0 ± 6.6	28.85 *	<0.001
Experimental Group	194	76.7	58	22.9	1	0.4			45.0 ± 8.8		

χ^2^—Pearson chi-square test, *—statistical significance.

**Table 4 vaccines-11-00727-t004:** Impact of the educational intervention on knowledge and concern about COVID-19 vaccination in the experimental group after adjusting for possible confounders.

	Knowledge Score			Concern Score		
	B (SE)	t-Value	*p*-Value	B (SE)	t-Value	*p*-Value
Study Group (experimental = 1)	17.09 (1.67)	10.24	<0.001 *	119.45 (0.72)	−27.00	<0.001 *
Gender (female = 1)	0.84 (1.65)	0.51	0.61	0.65 (0.71)	0.91	0.36
Anti-inflammatory drugs (yes = 1)	5.10 (2.36)	2.16	0.03 *	−1.83 (1.02)	−1.79	0.08
Immune-suppressive drugs (yes = 1)	4.45 (2.58)	1.72	0.09	−3.09 ((1.12)	−2.76	0.006 *
Constant	54.87 (1.25)	43.81	<0.001	65.11 (0.54)	119.91	<0.001

B—coefficient of determination, SE—standard error, *—statistical significance.

## Data Availability

Most of the data supporting our findings are contained within the manuscript, and all others, excluding identifying/confidential patient data, will be shared upon request by contacting the corresponding author (Mostafa Abolfotouh, mabolfotouh@gmail.com).

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
