# Peer review of "Impact of a Video-Based Educational Intervention on the Levels of Knowledge and Concerns about COVID-19 Vaccination"

_vaccines, 2023, doi:10.3390/vaccines11040727_

Round 1

Reviewer 1 Report

Thanks for the opportunity to review this paper. The situation and background of the research problem is well stated and the research method is clear. However, I am not convinced it is value-added research to practice and research. The topic of this paper is "Impact of a video-based educational intervention on the levels of knowledge and concerns about Covid-19 vaccination". But the study design is just to verify whether people who watched the COVID 19 education videos (experimental group)actually learned. It is too obvious and not interesting at all. Any healthy adult will learn immediately. I suggest the authors revise the research design to compare people learning between who watched the video (experimental group) and people who only access the traditional reading/learning (control group). So that we can see how effective the video-based educational intervention is comparing to traditional learning. 

Author Response

Thank you. The aim of the study was rewritten, putting your suggestion into consideration, as follows “This study aimed to assess the impact of a video-based educational session, in comparison to the traditional learning from the national health authorities, on the Saudi public's levels of knowledge and concerns about COVID-19 vaccination”.[lines 86-88].

One of the limitations of this study design was that the posttest was carried out immediately after the video. Although this methodology tests the amount of knowledge and concern acquired after the intervention session, it fails to evaluate the long-term retention of such outcomes. [strengths and limitations, lines 315-318]

Reviewer 2 Report

Insert “Personal and COVID-19-related characteristics” in material and method, please. It is the groups description. It’ no an objective of the study.   Explain the results in discussion, please. Why the impact of educational intervention on concern about COVID-19 vaccination was the highest in the mistrust domain? Why the impact of educational intervention was the highest on rumors? Why the history of taking anti-inflammatory drugs was positively associated while history of taking immune suppressive drugs was negatively associated with the percentage mean score of the overall concern about COVID-19 vaccination?

Author Response

Reviewer # 2

Insert “Personal and COVID-19-related characteristics” in material and method, please. It is the groups description.

“A subsection title was added “study participants” in the methods section [line 100]

Explain the results in discussion, please.

Why the impact of educational intervention on concern about COVID-19 vaccination was the highest in the mistrust domain?

A paragraph was added to explain the findings as follows “In a previous study on recovered Covid-19 patients, mistrust was the main attitudinal barrier of receiving the vaccine, where the majority of participants were not sure that vaccination would stop contracting serious infectious diseases. However, including clips that were structured specifically to focus on COVID-19 vaccine-related rumors and the scientific disclaimers of each of these rumors based on evidence would justify such impact.” [discussion, lines 265-270]

Why the impact of educational intervention was the highest on rumors?

A statement was added to explain this issue “This impact could be attributed to the clips that were structured specially to cover COVID-19 vaccine-related rumors and the scientific disclaimers of each of these rumors based on evidence”. [lines 279-282]

Why was the history of taking anti-inflammatory drugs positively associated while the history of taking immune suppressive drugs was negatively associated with the percentage mean score of the overall concern about COVID-19 vaccination?

The statement on drug therapy and the impact of the intervention was rewritten as follows “In our study, both history of taking anti-inflammatory drugs and immune suppressive drugs impacted positively on the overall knowledge and concern about COVID-19 vaccination, by raising the levels of knowledge and lowering the levels of concern about vaccination among drug takers. This finding could be justified by the fact that participants on multiple drug therapy had more concern and hesitance about taking the COVID-19 vaccine as it might deteriorate the prognosis of their condition [43,45].” [discussion, lines 289-294]. Meanwhile, the statement was rewritten in the results section as follows “A history of anti-inflammatory drug use was associated with a significantly higher percentage mean score of overall knowledge about COVID-19 vaccination (t = 2.16, p = 0.03), while the history of immune-suppressive drug use was associated with a significantly lower percentage mean score of overall concern about COVID-19 vaccination (t = −2.76, p = 0.006).”[Results, lines 241-245]

Reviewer 3 Report

This paper examines at least two important issues— (1) how to potentially improve people's understanding of Covid-19 and thus, we hope, their likelihood of being vaccinated and (2) then, more specifically, how such attempts at improvement might especially affect the Saudi Arabian population.

Not knowing much about these issues specifically in Saudi, I did a quick search to see what was already known about issues in that country of Covid-19 and Covid-19 vaccination knowledge, hesitancy, and related issues.  I was surprised to find a number of studies not addressed or even mentioned in the current paper (e.g., Al-Hanawi et al, 2020; Bazaid, et al., 2020; Al-Hazmi et al, 2021; Alanazi and Bahjri, 2022, and others).  Thus, my major concern is that the authors of the current study, in the introduction and discussion, address their results and conclusions and approaches in the context of these many other studies done in Saudi. It is difficult to evaluate their study without such contextualization, comparison, and critique.  For example, how did these previous studies affect their decision to do their study?  Did things change in Saudi and in results over time?  Etc.

Additionally, the authors should address more effectively why they did only a post-test and not also a pret-test for comparison, which it seems would be a stronger study.  They make an attempt, I think, to explain this in lines 276-282, but I don’t follow their logic.  A second thing the authors should address is the possible impact of having significantly more females in the experimental group, especially after they note in the paper that gender has been shown previously to affect in Saudi Arabia knowledge and understanding of Covid-19.

Finally, the English writing and grammar needs improvement, which is an easily addressable problem with a good editor.  Most of the issues are minor grammatical ones, but there are several that make the story the authors are telling unclear or confusing.  For example, the use of the word ‘explored’ in Figure 1 and its explanation in the text, and the use of the word ‘compression’ in the abstract and elsewhere, which I think is instead supposed to be ‘comparison’.

Author Response

Reviewer # 3

This paper examines at least two important issues— (1) how to potentially improve people's understanding of Covid-19 and thus, we hope, their likelihood of being vaccinated and (2) then, more specifically, how such attempts at improvement might especially affect the Saudi Arabian population.

Not knowing much about these issues specifically in Saudi, I did a quick search to see what was already known about issues in that country of Covid-19 and Covid-19 vaccination knowledge, hesitancy, and related issues.  I was surprised to find a number of studies not addressed or even mentioned in the current paper (e.g., Al-Hanawi et al, 2020; Bazaid et al., 2020; Al-Hazmi et al, 2021; Alanazi and Bahjri, 2022, and others).  Thus, my major concern is that the authors of the current study, in the introduction and discussion, address their results and conclusions, and approaches in the context of these many other studies done in Saudi. It is difficult to evaluate their study without such contextualization, comparison, and critique.  For example, how did these previous studies affect their decision to do their study?  Did things change in Saudi and in results over time?  Etc.

Thank you. All the studies suggested by the reviewer were included in the background and used in discussing our results. Moreover, intervention studies related to Covid-19 vaccination were discussed and compared to our study’s results. [Introduction, lines 80-84]

Additionally, the authors should address more effectively why they did only a post-test and not also a pre-test for comparison, which it seems would be a stronger study.  They make an attempt, I think, to explain this in lines 276-282, but I don’t follow their logic. 

 Agree. We included the absence of pre-intervention data as one of the limitations of this study. [Strengths and limitations lines 315-318]

A second thing the authors should address is the possible impact of having significantly more females in the experimental group, especially after they note in the paper that gender has been shown previously to affect Saudi Arabia's knowledge and understanding of Covid-19.

Agree. A new paragraph was added to highlight this issue [discussion, lines 298-303]

Finally, the English writing and grammar needs improvement, which is an easily addressable problem with a good editor.  Most of the issues are minor grammatical ones, but there are several that make the story the authors are telling unclear or confusing.  For example, the use of the word ‘explored’ in Figure 1 and its explanation in the text, and the use of the word ‘compression’ in the abstract and elsewhere, which I think is instead supposed to be ‘comparison’.

Thank you. The manuscript was English edited by the journal’s editing services

Round 2

Reviewer 3 Report

The authors have adequately addressed all of my previous concerns.  Thank you.